# Lobular Carcinoma of the Breast: A Comprehensive Review with Translational Insights

**DOI:** 10.3390/cancers15225491

**Published:** 2023-11-20

**Authors:** Harsh Batra, Jason Aboudi Mouabbi, Qingqing Ding, Aysegul A. Sahin, Maria Gabriela Raso

**Affiliations:** 1Department of Translational Molecular Pathology, The University of Texas MD Anderson Cancer Center, Houston, TX 77030, USA; hbatra@mdanderson.org; 2Department of Breast Medical Oncology, The University of Texas MD Anderson Cancer Center, Houston, TX 77030, USA; jamouabbi@mdanderson.org; 3Department of Pathology, The University of Texas MD Anderson Cancer Center, Houston, TX 77030, USA; qqding@mdanderson.org (Q.D.); asahin@mdanderson.org (A.A.S.)

**Keywords:** ILC, breast carcinoma, invasive lobular carcinoma, translational medicine, tumor microenvironment, hormone receptor-positive breast cancers

## Abstract

**Simple Summary:**

Invasive lobular carcinoma (ILC), the second most common type of breast cancer, is a distinct entity. Despite their unique biology and clinical course, lobular carcinomas have been researched in the broad category of hormone receptor positive breast cancers and these too have been predominated by invasive ductal cancers. With advancements in digital techniques, clinical research targeting ILC specifically is imperative in this era of targeted therapy. In this review, we highlighted the most important recent and relevant developments that have happened in the ILC domain, including histology and integrating advances in genomics and the overall translational medicine aspects of ILC.

**Abstract:**

The second most common breast carcinoma, invasive lobular carcinoma, accounts for approximately 15% of tumors of breast origin. Its incidence has increased in recent times due in part to hormone replacement therapy and improvement in diagnostic modalities. Although believed to arise from the same cell type as their ductal counterpart, invasive lobular carcinomas (ILCs) are a distinct entity with different regulating genetic pathways, characteristic histologies, and different biology. The features most unique to lobular carcinomas include loss of E-Cadherin leading to discohesion and formation of a characteristic single file pattern on histology. Because most of these tumors exhibit estrogen receptor positivity and Her2 neu negativity, endocrine therapy has predominated to treat these tumors. However novel treatments like CDK4/6 inhibitors have shown importance and antibody drug conjugates may be instrumental considering newer categories of Her 2 Low breast tumors. In this narrative review, we explore multiple pathological aspects and translational features of this unique entity. In addition, due to advancement in technologies like spatial transcriptomics and other hi-plex technologies, we have tried to enlist upon the characteristics of the tumor microenvironment and the latest associated findings to better understand the new prospective therapeutic options in the current era of personalized treatment.

## 1. Introduction

Invasive lobular carcinoma (ILC) ranks as the second most common type of breast carcinoma, representing approximately 15% of all breast cancer cases [1,2]. The incidence of this “special” subtype has increased in recent times due in part to hormone replacement therapy and improvement in diagnostic modalities [3,4]. Compared to IDCs, these are multicentric, multifocal, and bilateral, and the risk factors include alcohol consumption, previous hormone replacement therapy in postmenopausal women, elderly gravida, and previous family history [5,6,7]. Genetic and geographic factors seem to play a role as a higher incidence is seen in the western population when compared to the rest of the world [1]. Although believed to arise from the same cell type as their ductal counterpart, invasive lobular carcinomas (ILCs) are a distinct entity with different regulating genetic pathways, characteristic histologies, and different biology. Loss of E-cadherin, a key feature of this tumor type is due to CDH1 mutation, which occurs simultaneously with heterozygous deletion in chromosome 16q in majority of the cases [8,9,10]. Classic ILC is ER and PR positive and Her2 negative [11], while higher Her2 positivity is seen in the pleomorphic variant (30 to 80%) [3,12,13]. Although these tumors have a better clinical prognosis initially, they tend to recur at a later age. Multicentricity [14] with peculiar metastatic sites differs from its ductal counterpart [15]. Metastasis to the ovaries, bones, leptomeninges, digestive tract, orbital tissue, and skin is more frequent in ILC than IDC [11,16,17,18,19]. Despite being a distinct type of breast cancer, ILC-specific research studies have often been overshadowed by their ductal counterparts, which can be gauged from the fact that most major drug trials have been designed with breast cancer cases as a broad blanket category and with the case cohort being dominated by ductal counterparts. In an era of targeted therapy, it becomes imperative to consider this entity solely. The aim of this review is to provide an overview of ILC in terms of its histology, molecular characteristics, its microenvironment, and recent advances.

## 2. Pathogenesis

Arising in lobules and terminal ducts, invasive lobular cancer was first described by Foote and Stewart [20]. Initially thought to arise with an accompanying precursor lesion like atypical lobular hyperplasia (ALH) and lobular carcinoma in situ (LCIS), these are now known to be nonobligate lesions, and ILC can occur exclusive of these lesions [21]. Inactivating mutation of CDH1 (bi-allelic, homozygous deletions, and epigenetic mechanisms such as promotor methylation) are the most common causing E-Cadherin loss and the loss of cohesion of cells [8]. Further mutations like TP53 and ERBB2 lead to de-differentiation and to the development of various histological types of ILC. Rarely, there can be inactivation of certain other tumor suppressor genes like αE-catenin, which can lead to the development of this tumor type [22].

## 3. ILC Histological Subtypes

Classic ILC (Figure 1): Up to 55% of the cases belong to the classical subtype, initially described by Martinez and Azzopardi, showing small, discohesive tumor cells in a single file pattern with little stromal reaction. The tumor cells may encircle the mammary duct in a target-like fashion, which is known as peri-parenchymal streaming, a characteristic feature [23].

Alveolar ILC (Figure 2): This histological subtype shows tumor cells lying in loose alveolar pattern (usually more than 20 tumor cells), with intervening thin collagen bands, as described in 1979 by Martinez and Azzopardi [23]. Hypothesized to be an interim phase preceding the appearance of discohesive tumor cells, they are low-grade tumors with a low proliferative index [23]. Solid ILC: Described in 1975 by Fechner et al. [24], this subtype has an overall poor prognosis and is associated with a higher proliferative rate. Histologically, it is characterized by tumor cells with lobular cytomorphology growing in solid sheets. Three recent separate studies by Christgen et al. [25], Rakha et al. [26], and Motanagh et al. [27] have also described a similar subtype, which can show tumor cells lying in solid sheets admixed with tumor cells showing fibrovascular cores, thus describing them as a solid papillary variant of ILC.

Trabecular ILC (Figure 2): This subtype shows tumor cells arranged in in “broad bands” or “trabeculae”, which can be up to three cells thick. These are usually low grade with a low proliferative index [23].

Nonclassic ILC: These account for the minority of cases of ILC but are characterized by a high grade and aggressiveness. *Pleomorphic ILC* (Figure 3): These are characterized by tumor nuclei which are three to four times the size of a lymphocyte (nuclear diameter ≥ 18 µm) [28], have moderate to marked anisocytosis with the variable presence of prominent nucleoli. Although highly proliferative, high grade in nature, with a higher number of mutations (DNA CN alterations, ER negativity, Her2 positivity, higher rate of TP53 mutation, IRS2 and IGFR1 mutations, FER Kinase expression, mutations in DNA methylation, ESRRA mutation, especially in triple negative ILCs), with higher scores on conventional gene profiling assays compared to classic ILC [12,29,30,31,32,33,34,35,36,37,38,39,40,41], their prognosis has been debatable, with the works of Eusebi et al., Weidner et al., and Bentz et al. [42,43,44] conveying a poor prognosis and recent studies reporting a five-year survival rate of about 77% when treated with modern regimens [45,46]. However, when adjusted for Her-2 status, the hormone receptor-negative/Her-2 positive pleomorphic ILC has a poorer prognosis [46].

*Histiocytoid ILC* (Figure 4): Initially described by Hood et al. [47] in 1973, these cases display discohesive tumor cells with some arranged in a single file pattern, with nuclei with less pleomorphism, and having clear to foamy cytoplasm. Negative for ER and PR, these are positive for AR [48,49]. Histiocytic markers like PGM1 are positive in these tumors [50]. It is important to highlight that this specific subtype has faced diagnostic challenges due to its similarity to apocrine carcinoma (showing GCDFP-15 IHC positivity) and its co-occurrence with pleomorphic ILC, leading some to consider it a variant of pleomorphic ILC [51,52]. The crucial factor in accurate diagnosis [51] lies in recognizing this rare phenotype through careful histological examination and the application of IHC markers (such as E-Cadherin and cytokeratin).

*ILC with signet ring cells* (Figure 5): These ILCs show tumor cells showing intracytoplasmic vacuoles with mucin and nuclei pushed to the periphery without the presence of extracellular mucin [51,52]. Ultrastructural studies show microvilli in the membrane of these vacuoles [53].

## 4. Immunohistochemistry

Although the tumor is mostly characteristic with respect to histomorphology, a recent study still showed that most pathologists use E-cadherin IHC for the diagnosis of ILCs. Loss of E-cadherin on IHC (Figure 6) is a characteristic of these tumors, and it is better to perform this IHC in cases of ILC to avoid misdiagnoses. A recent multi-institutional study where thirty-five pathologists from nine different countries took part, evaluated the concordance rate of reporting invasive lobular carcinoma. The study found that although there was a modest correlation upon the initial diagnosis, application of E-cadherin IHC to ILC cases led to near-perfect concordance of reporting amongst observers [54,55].

Although less common, the use of p120 catenin IHC (membranous expression) (Figure 7) has also been found to be useful in lobular carcinoma cases. It is a tyrosine kinase substrate and is helpful in differentiating especially the early precursor lesions like ALH and LCIS when a differential of low-grade DCIS is concerned [56]. The most common IHC markers used in breast cancer cases include hormone receptors, Her2Neu, and the Ki67 index. The molecular subtyping into Luminal A (ER+ or PgR+, Her2−), Luminal B (ER+ or PgR+, Her2+), Basal-like (triple negative; ER−, PgR−, Her2−), HER2/Neu enriched (ER− and PgR−, Her2+) is also based on the receptor status [57,58].

Hormone receptor positivity (ER or PR positivity) is seen in up to 90% cases of ILC [59]. Her2 overexpression constitutes about 13% of the cases. Triple negative ILCs range from 2 to 9% of cases, and the majority of these have androgen receptor positivity [28,60,61,62,63]. Her-2 low breast cancers are a relatively new and controversial concept in breast oncology. Antibody drug conjugates-based treatment regimens have been implied in conferring a relatively response-free and progression-free survival in Her-2 “low” (IHC 1+ or 2 with negative ISH) breast tumors [63,64]. Two large studies studying the prevalence of Her-2 “low” status have reported more than half of diagnosed cases of ILC are Her-2 “low” and these cases had a worse disease-free survival [DFS] [65,66]. The study by Van Balaen et al. [66] additionally found that amongst the HER-2 low cases, Her-2 1+ cases had a worse DFS compared to the Her 2 “0” cases and that Her-2 2+ cases had a worse overall survival. The role of the newer antibody drug conjugates thus needs to be explored, particularly in ILCs as they have a higher prevalence of such cases and might benefit from this therapy.

## 5. Molecular Alterations in Invasive Lobular Carcinoma

The most common driver mutations in ILC are CDH1, PIK3CA, FOXA1, PTEN, FGFR2, ERBB2, FGFR2 and ERBB3 [8,67,68]. In this review we summarized key mutations which have translational importance.

### 5.1. CDH1 Mutations

CDH1 mutations are the most frequent mutations seen in ILC (in up to 65% of cases) [8,69,70]. E-cadherin, a 120 kda is a transmembrane glycoprotein transcribed by the CDH1 gene. It has a highly conserved intracellular domain and a glycosylated extracellular domain. The extracellular domain binds Ca^2+^ and activates it, whereas it forbids catenin from having downstream effects by binding alpha, beta, gamma, and p120 catenins to its actin cytoskeleton. Loss of e-cadherin (loss of 16q) causes cytoplasmic accumulation of p120 catenin, which interacts with various effector molecules and pathways (e.g., Rho/Rock signaling pathway), thus causing anoikis resistance and tumor progression [70,71,72]. Loss of the CDH1 gene is central to lobular carcinoma pathogenesis and iterates the important concept of *synthetic lethality (in terms of its translational importance), which occurs when there are multiple combined deficiencies in the expression of two or more genes leading to cell death* [73]. The ROSALINE trial (NCT04551495) and the ROLo trial (NCT03620643) are currently investigating the use of ROS1 inhibitors in CDH1-deficient tumors [74,75].

### 5.2. Mutations of the PI3K/Akt Pathway

According to the TCGA data, mutations related to the PI3K/Akt signaling pathway are the second most common in ILC (35–48%). Studies have found that acquired resistance to estrogen-deprivation therapies is linked to PI3K/AKT/mTOR constitutive activation [76,77,78]. Its translational importance is evidenced by two large randomized trials, BOLERO-2 and TAMRAD, which found a better progression-free survival of HR+ breast cancer with the addition of a TORC1 inhibitor, Everolimus, in patients who had earlier received the aromatase inhibitor Exmestane and showed an initial response but progressed later in the course of therapy [79,80,81]. Recently, a phase 3, randomized, double-blind trial included a study population of pre-, peri-, and post-menopausal women and men with hormone receptor-positive, human epidermal growth factor receptor 2–negative advanced breast cancer who had had a relapse or disease progression during or after treatment with an aromatase inhibitor, with or without previous cyclin-dependent kinase 4 and 6 (CDK4/6) inhibitor therapy. Capivasertib plus fulvestrant intervention was studied, and progression-free survival was assessed in patients with mutated AKT pathway altered and in the normal population. It was shown that Capivasertib–fulvestrant therapy resulted in significantly longer progression-free survival than treatment with fulvestrant alone [82].

### 5.3. IGF1 Pathway

The IGF1 pathway is known for its association with breast cancer progression. Binding of IGF to its receptor IGF1R further leads to activation of the PI3K and MAPK pathways. Higher IGF1/2 expression and its receptor activation is seen in ILCs compared to ductal carcinomas [83]. E-cadherin loss increases the availability of the IGF1R receptor, thus causing increased IGF1, IGF2, and insulin signaling and, hence, leading to increased ligand binding. This ultimately leads to tumor cell motility and invasion. The IGF1R axis is a plausible setting for ILC treatment using IGF1R, PI3K, Akt, and MEK inhibitors [84].

### 5.4. ERBB2 Mutations

ERBB2 amplifications, which exert their oncogenic effect by heterodimerization to HER3, are frequently found in ILC (in up to 15%) [12,41,85,86,87]. ERBB2 mutations, which can be nonamplified, or somatic mutations, which drive the oncogenesis by tyrosine kinase activation or dimerization of the HER2 domain, vary in frequency from 4 to 6% in ILC according to studies [67,88,89]. These mutations are found in tumors with higher grade and higher reported metastasis. While the CDH1 mutation is mutually exclusive of ERBB2 mutation, it has been shown that tumors possessing both the mutations have a comparatively worse prognosis and higher recurrence rates [87,90]. HER 2 mutated tumors (nonamplified and somatic mutations) show an early relapse and an increased resistance to endocrine therapy. The pleomorphic variant of ILC has been shown to harbor these mutations in the majority of cases with this specific histology (up to 50% cases of pleomorphic ILC) [12]. Recent trials have shown that the addition of Neratinib to Fulvestrant and Trastuzumab is effective in these HER mutated tumors with better antitumor activity [91,92].

### 5.5. Fibroblast Growth Factor Signaling

Fibroblast growth factors (FGFs) have a central role in regulating cell growth and angiogenesis. The fibroblast growth factor signaling pathway has a distinct part to play in ILC tumor biology. Studies have shown that FGFR1 acts as an initial amplicon in ILC genetics, and inhibition of cell growth in cell lines was observed when FGFR1 was inhibited [93]. The receptor is associated with poor prognosis, disease recurrence, and resistance to endocrine therapy and to CDK4/6 inhibitors [94]. The MONALEESA-2 trial showed that patients with *FGFR1* amplification had rapid disease progression compared to patients without the mutation. Presently, an active phase Ib clinical trial is evaluating Fulvestrant, Palbociclib, and Erdafitinib (FGFR inhibitor) in endocrine-resistant ER^+^ HER2^−^ metastatic breast cancer patients with amplified FGFR genes (NCT03238196) [95].

### 5.6. Endocrine Resistance and Fox A1 Amplification

FOXA1 is a transcription modulator which influences transcription for the estrogen receptor. It can bind and demethylate condensed chromatin, removing the H1 linker [67,96]. Recent studies have shown that FoxA1 mutations, which are seen in 7–9% [67] of ILC, are involved in metastasis and endocrine resistance [96,97]. A recent study using preclinical models has also found that FOXA1 drives a distinct chromatin state in ILCs where FOXA1 auto induces itself using super-enhancers, building a feed forward circuit with conjecture action with transcription factors like GATA3 and ER [98]. Also, FOXA1 has been shown to play a role in tamoxifen resistance in ILC as it helps the tumor retain the ER binding sites which tamoxifen acts on [98]. These plausible mechanisms could constitute the basis for targeted drug therapy, for example, the RET Inhibitor Pralsetinib as a direct ER target gene.

### 5.7. ESR 1 Mutations

The estrogen receptor, which acts as ligand-dependent transcription factor, goes to a ligand-independent state with the mutation of its gene ESR1 [99]. Studies have found that these mutations are frequently found in metastatic settings in ER+/Her 2 negative tumors, especially following aromatase inhibitor therapy [100,101]. Various studies report a frequency rate of up to 26% in ILCs [102,103,104]. Since the approach to tumors with this mutation requires a different endocrine therapy approach, recognizing these is imperative in a metastatic ILC setting. It has also been found in a study by Desmedt et al. that these mutations are found at a much higher frequency in metastatic settings compared to the primary tumor, thus showcasing their possible de novo nature; however, they did not rule out that a minor subclone in the primary tumor might have had these mutations to begin with [105]. The use of selective estrogen receptor degraders (SERD) like fluvestrant (with or without Palbociclib) has found success but with limitations in ER+/Her-2 negative breast cancers. Recent results from the randomized phase III EMERALD trial have shown greater efficacy and progression-free survival rates with the use of Elacestrant as compared to standard endocrine therapies [106].

### 5.8. APOBEC and Tumor Mutational Burden

Apolipoprotein B mRNA-editing enzyme catalytic polypeptides (APOBEC) are a group of cytidine deaminases, which degrade viral genomes via cytosine deamination [107,108]. They cause cytosine-to-thymine (C-to-T) mutations driving tumorigenesis in humans. High tumor mutational burden (TMB) is seen in 16% of metastatic ILC (mILC) and 4.7% of primary ILC [109,110]. APOBEC signatures have mostly been studied with respect to TMB, which is considered a biomarker when evaluating immune checkpoint inhibitor therapy response. Studies by Chumsri et al. [111] and Pareja et al. [70] have showcased higher APOBEC mutational signatures in ILC, specifically in metastatic ILCs. Although these hypotheses in the context of ILC are still naïve, these results can very well pave the way to build trials testing ICI in a subset of ILC cases.

### 5.9. Germline Mutations

Germline mutations in ATM, PALB2, CHEK2 (moderate risk) and CDH1, BRCA2 (high risk) are associated with risk of ILC [112,113,114]. Although the frequency of germline mutations is lesser compared to the ductal counterpart, mutations such as BRCA2 and CHEK2, which are involved in near-equal frequencies and increase the risk of developing ILC, iterate the fact that germline mutation testing should be done in ILC cases. The CDH1 gene is highly associated with germline lobular breast cancer. Its germline mutation frequency is estimated to be around 3–4%. However, this genetic risk is insufficient to clarify the cancer predisposition for lobular breast cancer. So, a recent clinical trial (LobularCard) is searching for novel genetic factors, especially in cases with early onset of lobular breast cancer. The trial is researching a set of 113 previously known breast carcinoma genes (NCT05410951) [115].

## 6. Gene Expression Profiling Tests

The Oncotype DX 21-gene assay is a quantitative reverse-transcriptase polymerase chain reaction assay that has a prognostic role in terms of recurrence and effectiveness of chemotherapy in early HR+HER2-negative BCA. The very initial studies [116] laid out three cutoff categories with a score which ranges from 0 to 100, categorized by recurrence probability into three categories: low risk (RS < 18), intermediate risk (RS, 18 to 30), and high risk (RS ≥ 31). Few studies exploring the utility of this assay in ILCs have used slightly relaxed criteria [117,118], categorizing low risk as RS < 11. Almost all the studies iterated that the use of this assay for ILC cases is mostly associated with low to intermediate RS, with very few percentages of cases being high RS (0–8%) and showing a poorer response to adjuvant chemotherapy [45,117,118,119,120,121,122,123,124,125,126]. However, a recent study which evaluated the use of this assay using cases from the national cancer database, out of which 20% of cases were ILC cases, found that although ILC cases had a genomically low risk, the clinical risk was higher [127]. This discordant nature, especially in ILC, translates to difficulty in deciding a specific treatment regimen and iterates the importance of ILC-specific prediction tools.

The 70 gene MammaPrint assay is a microarray-based test which analyses the expression of 70 MammaPrint genes [128], assessing recurrence risk in early-stage breast cancer [129,130,131], and also helps in treatment decisions, viz., adjuvant vs. neoadjuvant therapy. A study by Beumer et al. [132] found the assay to predict adverse prognosis for distant metastasis-free survival (DMFS) and overall survival in ILC. Similarly, the MINDACT [133] trial, which had 9% of cases as ILC, found a prognostic role of the assay when seen in ILC, and with the latest release of results of the study, it has also been found that this genomic assay is able to help make the decision of including or not including chemotherapy in an ILC case when the scores are seen along with the age and lymph node status of the patient. Two national cancer data-based analyses have iterated that the assay does have prognostic significance and that ILC seems to have more discordance in being genomically low risk but clinically high risk, but the predictive capability in terms of chemotherapy treatment is questionable [134,135]. Recently, a study which evaluated this assay has reiterated that ILC cases, especially in patients under 50 years of age, are more likely to fall into the clinically high and genomically low risks groups, and, thus, it limits the use of such an assay, which, although robust, is not tailored specifically to ILC cases [135].

Endopredict (EPclin) is an RNA-based genetic score which assess the prognosis in early-stage, ER-positive, and HER2-negative breast cancer, using a panel of twelve genes (eight cancer-related genes—*UBE2C*, *DHCR7*, *BIRC5*, *RBBP8*, *IL6ST*, *AZGP1*, *MGP*, and *STC2*—and three normalization genes—*CALM2*, *OAZ1*, and *RPL37A* and one control gene) *in conjuncture with clinical status, tumor size and node status, and it generates a* score predicting distant recurrence likelihood. In studies which included cohorts from ATAC, ABCSG-6, and ABCSG-8 trials, EPclin shows good prognostic utility in ILC and was found not to be limited by node status or time point of recurrence of the disease [136]. Also, this assay is more cost-effective as it can be reproduced locally rather than determining the results through a central lab and can be used for both early and late time points of distant recurrence.

PAM50 Prediction Analysis of Microarray uses a 50-gene assay [137], and a study by a Danish group of researchers has found significant prognostic value (10-year disease recurrence rates) for this assay in ILC cases [138].

The Breast Cancer Index (BCI) utilizes an algorithm combining the *HOXB13:IL17BR* ratio (H/I) and the Molecular Grade Index (MGI), increasing its utility compared to the two markers when used [139]. Its utility has been explored in a single dedicated ILC study consisting of 307 cases, and the results indicate that BCI is an important independent predictor for risk stratification in ILC and that it also has a role in stratifying cases which might need a de-escalation of treatment if they are classified as low risk using this index [140].

The Genomic Grade Index (GGI) is based on the average expression levels of 97 genes associated with histologic grade in breast cancers, with the index being directly proportional to the grade of the tumor [141]. Although this assay has a role in predicting prognosis in ER+ IDC in earlier studies, there has only been a single study comparing its utility particularly to highlight the high-grade ILC cases, and a significant prognostic value has been found in these cases [142].

Though the above gene expression assays have been the most prevalent in terms of being breast cancer specific, with the above, we can see that regarding data, though consistent to an extent, there exists a discrepancy in terms of a firm conclusion as to whether any of these above assays, singly or combined, offer a true and dedicated prognostication in ILC. This is because the studies have combined breast carcinomas as a single entity and were predominated by IDC cases, and lobular carcinomas exhibit a genetic heterogeneity amongst themselves, as described in a study by Mcreed et al. [9]. So, a dedicated ILC prognostication tool should exist to tackle this unique carcinoma in a targeted way.

### Lobsig

This ‘signature’ was developed using TCGA and METABRIC cohorts, merging DNA copy number data with gene expression data to identify 194 genes with prognostic value in ILC. LobSig outperformed other widely used breast cancer prognostic tools, such as the Nottingham Prognostic Index, PAM50, OncotypeDx, and Genomic Grade Index. While further validation is needed, and its applicability to various settings requires testing, LobSig stands as the first dedicated signature developed specifically for ILC to date [143].

## 7. Tumor Microenvironment in ILC

Being unique in its biology compared to breast carcinoma NST, ILC has conventionally been thought to have a quiescent immune landscape. So, despite being distinct, there have been limited studies, which are increasing now, targeting ILC specifically.

**Tumor infiltrating lymphocytes (TILs)**: The role of TILs has become important in recent years due to their association with immune checkpoint blocks like PDL1 and targeted therapies [144]. Recent studies by Tille et al. [145] and Desmedt et al. [146] focusing on TILs on the ILC microenvironment have shown that although ILC has a lesser infiltration of TILs than IDC-NST, a subset of ILCs have an immune-rich microenvironment that impact the behavior of the tumor. TILs were found to be associated with young age, larger tumor size, nodal involvement, HER2 amplification, multinucleation, and prominent nucleoli [145,146]. The study by Desmedt et al. [146] found *ERBB3* mutations to be associated with a lower TIL count and tumors with *TP53*, *ARID1A*, and *BRCA2* mutations to be associated with higher TILs. It has also been found that ILC has low intratumoral CD8 content, and, thus, most of the infiltration is in the stroma, hence, decreasing cancer immunity in ILCs [146]. The infiltration of sTIL in relation to checkpoint inhibitors like PD1 and PD-L1 has been an area of interest because of translational importance. Thompson et al. studied PD-L1 expression in ILCs and found that a subset of ILC cases did exhibit PD-L1 positivity but were not associated statistically with immune cells infiltrate quantity, histological grade, or receptor status [147]. Interesting results have emerged from the recent phase II GELATO trial, which explored the utility of Atezolizumab (anti-PD-L1 monoclonal antibody) for patients with advanced lobular breast cancer. In a small cohort of 23 patients with advanced lobular breast cancer, a combination of induction by low-dose carboplatin and Atezolizumab after two weeks of treatment of carboplatin was studied. Although the majority of the patients did not achieve a clinical benefit, six patients did achieve partial response or stable disease with increased levels of immune cell infiltration (CD8 positive immune cells), immune checkpoint expression, and exhausted T cells [148]. Although these can be seen as negative results, they hold a translational importance, especially in “immune-cold” tumors such as ILC. It also emphasizes that future clinical trials should include more patients from the ILC subset that are immunogenically more active and have a higher likelihood of obtaining benefit from immune therapies.

**Macrophages**: Being immune cold [145,146,147,148], there is an imperative need to study the tumor microenvironment (TME) of ILC in detail and search for other targetable entities. It has been studied and accepted that mechanisms mediating immunosuppression by tumor-associated macrophages (TAMs) can act and influence mechanisms involved in immunosuppression, and, hence, they also present as potential targetable entities [149]. Macrophages in the TME are broadly classified as M1 and M2 based on their antitumorigenic and protumorigenic potential [150]. A study by Sayali et al. [151] showed higher stromal concentration of M2-like macrophages in ILC compared to IDC. These results, along with other studies [152,153], highlight a distinct TME in ILC, which shows that T cells have difficulty to get to the tumor effectively in an expanded macrophage environment in ILCs, and, thus, there is hampering of subsequent tumoricidal activity. More such studies are required for validation and to discover new therapies which can repolarize macrophages holding an importance in ILC treatment.

**Cancer-associated fibroblasts (CAFs)**: Cancer-associated fibroblasts form the majority of the tumor-associated stromal component. They have been subdivided in previous studies from CAF S1 TO CAF S4 by permutating and combining different possibilities of the associated markers (FAP, α-SMA, FSP1, PDGFRβ, CD29, and CAV1) expressed [154,155]. CAF-S1 (CD29^Medium^ FAP^Hi^ FSP1^Low-Hi^ αSMA^Hi^ PDGFRβ^Medium-Hi^ CAV1^Low^) has been shown to be immunosuppressive [154], and FAP^Positive^ cells have also been demonstrated to exert an immunosuppressive activity in breast cancer mouse models [156,157,158]. An IHC-based study by Koo et al. [159] showed high levels of FAP-α, FSP-1/S100A4, and PDGFR-β in the stroma, and expression of FAP-α and FSP-1/S100A4 was higher in both tumor and stromal cells compared to invasive ductal carcinoma. In-depth analysis studies using laser microdissection and gene expression have highlighted a stromal difference between ILC and IDC with analysis of *PAPPA*- and *IGF1*-realted genes involved in a paracrine signaling pathway; PAPP-A secretion has been shown to come from CAFs in ILC [160]. Recently, a study by our group also showed a differential expression of CAF expression in the tumor microenvironment between classic ILC versus pleomorphic ILC. Whereas, the alpha-SMA population (alone or co-expressed with other CAF markers) seemed to be the predominant population in classic ILC, cases of pleomorphic ILC showed increased proximity of S100 expressing CAFs to the tumor cells, thus showcasing the differences in the tumor microenvironment within the same tumor but different subtypes [161,162] Thus, the role of CAFs seems to be active and promising in the ILC TME. Although drugs targeting CAFs have not achieved a practical use yet, it can be hypothesized that therapies against intermediate molecules like PAPPA [160] and DPP [163,164] which act via an interplay in CAF biology by modulating immune TME, can have a potential translational implication.

**Cancer-associated adipocytes (CAAs):** Cancer-associated adipocytes are an emerging concept involved in cancer progression and hold a novel importance, especially in tumors of breast which have the adipocytic component as a part of their histology [165,166]. CAAs cause fatty acid release, regulate inflammatory mediators and other protumorigenic molecules, thus modulating the tumor microenvironment to aid tumor growth [167]. Comparative genomic analysis and gene expression studies have demonstrated different mechanisms of lipid metabolism in ILC versus IDC [30,168,169]. IHC-based studies have analyzed differential expression of lipid metabolism enzymes and have found that ILC showed a higher expression of hormone-sensitive lipase (HSL) and fatty acid binding protein (FABP4) and a lower expression of perilipin A, with carnitine palmitoyl transferase 1 (CPT-1) and acyl-CoA oxidase 1 expression being associated with a shorter disease-free survival in ILC [170]. A recent study by Desmedt et al. [171] has explored the association of CAA and neoadjuvant endocrine therapy with letrozole where they correlated adipocyte size, BMI, and antiproliferative response. The group concluded that a higher BMI and increased adipocyte size are associated with greater antiproliferative response to neoadjuvant therapy, thus conferring a positive prognostic response. Recent spatial transcriptomic studies have also iterated a differential expression of adipocyte tumor contact in ILCs and the association of such signatures with relapse, greater PIK3CA association, and worse outcomes [172]. Since CAA are differentially expressed in ILC compared to IDCs, further work needs to be done in this domain to pin on translational roles.

Recent genomic and transcriptomics studies have found a differential composition of the tumor microenvironment in ILCs compared to IDCs. Lobular carcinomas have a higher expression of *TFAP2B*, *SOCS2*, *NOSTRIN*, *THBS4*, *SCUBE2*, and *GDF9* and a lower expression of *CDCA4*, *PSMG1*, *LMOD1,* and *SLC7A5*. In the subset of immune-enriched ILC cases, there is a higher expression of CTLA4, LAG3 and PDL1. A higher expression of TROP2 is also seen in ILCs compared to luminal IDCs [173].

## 8. ILC Radiology Aspects

Mammography, which is a widespread screening tool in the detection of breast cancers, has its limitations in ILC owing to the diffuse nature of the tumor. The sensitivity varies widely from 30 to 83% [174,175]. Reported in up to 50% of cases, whenever detected, mammography shows a high-density spiculated mass or asymmetric densities in ILC cases [176,177].

Ultrasound, which is not a routinely employed technique in mass breast screening, has shown a sensitivity ranging from 66–98% [178,179] in detection of ILC and is used as an adjunct tool in the detection of ILC, especially in women with dense breasts [180]. When done, the most common presentation is of a hypoechoic mass with posterior acoustic shadowing [179].

The most common presentation of ILCs on MRI is irregular or spiculated margins or a non-mass lesion [181]. Magnetic resonance imaging (MRI) has been reported to reach near-perfect sensitivity rates (93–100%) in ILC detection and has higher accuracy rates while detecting synchronous and bilateral lesions [174,181,182,183,184]. However, an increase in sensitivity has the downside of low specificity and consequently higher false positive rates [174,184]. Thus, although highly sensitive, the potential of MRI as an assessment tool before breast conserving surgeries has its limitations in cases of ILC. Despite its limitations, it is an important tool of surveillance in ILCs as it helps improve the detection of early-stage breast cancers in younger patients [185].

Computerized tomography (CT) has been a widely used modality and [^18^F]2-fluoro-2-deoxy-D-glucose (18-FDG) PET CT is especially a commonly used radiological modality to assess metastasis in various tumors, including breast carcinomas [186]. Since the technique is based on tumor metabolism and subsequent detection, its detection rates are comparatively lower in ILC cases owing to the tumor’s quiescent nature and low metabolic activity [187]. This has led to research on other fluoride-tagged metabolism tracing utilizing technologies using PET CT, and two studies have shown some promising data pertaining to improved detection of ILC metastasis. A study by Tade et al. [188] has explored a synthetic amino acid analog (*Anti*-1-amino-3-^18^F-fluorocyclobutane-1-carboxylic acid or ^18^F-fluciclovine) in a small cohort of thirteen cases, out of which five were ILCs, and found improved detection rates both for IDC and ILC. Another study developed by Ulaner et al. explores estrogen receptor binding tracers like 16α-^18^F-fluoroestradiol (^18^F-FES) to better detect metastasis. The study found that estrogen-based traces fared better than the conventional ^18^F-FDG PET/CT for detection of metastases in ILC [189,190]. Both these studies have now been expanded to ongoing ILC-specific clinical trials, and results are yet to be concluded [191,192].

## 9. Current Treatment Approaches

### 9.1. Neoadjuvant Chemotherapy

A well-known approach in breast carcinoma treatment is the administraton of neoadjuvant chemotherapy [NACT] preoperatively due to the envisioned benefits like tumor downgrading and achieving breast conservation surgery. However, the decision to prescribe neoadjuvant chemotherapy is not guided by the tumor’s histological subtype [193]. In a recent largest-of-its-kind meta-analysis by Connor et al. [194] to compare the sensitivity of NACT between ILC (pooled cases = 7596) and IDC (pooled cases = 79,708), it was shown that ILC had three times lower overall pathological complete response (pCR) rates compared to IDC. Thus, the use of NACT in ILC seems to be limited, which may be due to the receptor status and patients might benefit more from adjuvant NAET, as shown by studies we outlined in Section 9.4.

### 9.2. Surgery

Owing to its diffuse and multifocal nature, 17–65% of patients undergo a second surgery following a conventional breast conservation surgery with wide local excision [59]. Moreover, a complete mastectomy with contralateral mastectomy, due to associated bilaterally, is known in ILCs [195]. However, there seems to be no evidence of long-term survival for mastectomy or bilateral mastectomy in ILC patients compared to breast conservation surgery with negative margins and in combination with radiotherapy [196].

Axillary management in breast cancer patients has two options: sentinel lymph node biopsy (SLNB) and axillary lymph node dissection. It has been shown in the ACOSOG Z0011 study that ALND has a similar survival benefit to SLNB in node negative, SLNB positive tumors (≤2 lymph node positive) [197,198]. This study had only 7% cases as ILC, and only 27 were randomized to receive SLNB and ALND. Lobular cancers, due to their biology and diffuse nature, are more notorious for metastasis in SLNB as well as ALND; however, recent studies have found that SLNB offers similar levels of outcomes as compared to ALND in ILC patients [199,200,201]. However, in spite of these data, the surgical management is tailored to each patient in cases of widespread disease. It is to note that in cases with age > 70 years and HR+/Her2 negative tumors with significant comorbidities, axillary dissection is not done.

### 9.3. Adjuvant Radiotherapy

Although the local disease control rates have been shown to be similar following radiotherapy between IDC and ILC [202], the technique is not recommended in ILC owing to its multifocal nature [203].

### 9.4. Endocrine Therapy and CDK4/6 Inhibitors

Endocrine therapy is the widely preferred therapeutic modality for hormone receptor-positive ILC [204,205]. Early-stage ILC has shown to have a favorable response to neoadjuvant endocrine therapy (NAET), showing volume reduction in the tumor within a few months of therapy [206]. Aromatase inhibitors take the lead over tamoxifen for ILC treatment, as seen in two phase 3 trials, Breast International Group (BIG) 1–98 trial and ABCSG-8 study, showing overall improved survival rates with aromatase inhibitors [207,208].

An NAET combination with CDK4/6 inhibitors might find its use in early ILC, with one trial still trying to test its efficacy [209], but the area where this combination has shown a breakthrough and is somewhat considered a first-line therapy is in HR+/Her2 negative carcinomas. The evidence for this comes from PALOMA-2 phase 3 trials [210] with a Palbociclib and Fulvestrant combination showing an overall better survival benefit. This finding has also been supported by two U.S. Food and Drug Administration pooled analyses, which showed a clear survival benefit of CDK4/6i combination therapies [211,212]. However, the benefit of this therapy seems to apply to all HR+ breast cancers as was seen in a recent observational retrospective study which utilized a histology-based approach to segregate the metastatic ILC cases and to see if there was a differential benefit of the therapy in ILC cases in particular. The study found that the addition of targeted therapy to endocrine therapy had similar benefits in hormone receptor-positive cancers irrespective of the histology [213]. This retrospective study, which compared the survival outcomes in metastatic HR+/Her 2 negative IDC vs. ILC treated with endocrine therapy plus CDK4/6i, showed that in terms of survival outcome, the benefit of the therapy [median PFS: 11.7 months (IDC) 14.5 months (ILC), PFS at 1 year, 5 years, and OS: 34.2 months (IDC vs. 34.6 months (ILC)] was similar in IDC and ILC. The three CDK4/6i currently being explored in trials are Abemaciclib (MONARCH 2, MONARCH3) [214], Palbociclib (PALOMA 2, PALOMA 3) [210], and Ribociclib (MONALEESA 2, MONALEESA 3, MONALEESA 7) [215,216]. A pooled analysis of phase 3 trials from 2013 to 2017 was performed by the U.S. FDA, evaluating the three CDK4/6 inhibitors. The evaluation was based on PFS, OS, and overall response rate [211]. The four arms of the study intervention were CDK4/6i+AI, CDK4/6i+Fulvesterant, Placebo+AI, and Placebo+Fulvesterant. The CDK4/6i addition had a benefit on the median PFS, but the drugs performed equally efficiently in the IDC and ILC cohorts. Another aspect that the trials evaluating CDK4/6i confer is that the three agents act in their unique ways different from each other. Ribociclib has a higher CDK4 inhibitory activity than CDK6 inhibition, and Abemaciclib has a higher CDK1 and CDK2 action [217]. Thus, CDK4/6i therapy does seem a better adjuvant approach compared to endocrine therapy alone, but the clinical and toxicity profile of each patient must be considered with these agents. This also iterates that there is a need to discover novel biomarkers and better imaging technologies and for exploration of role of ctDNA to carve out a personalized therapy plan.

An aromatase inhibitor is the preferred initial therapy for the first few years in ILC patients. However, there seems to be similar benefit in terms of disease-free survival of the addition of tamoxifen initially, followed by an aromatase inhibitor like Exmestane vs. Exmestane-only therapy in ER positive tumors (TEAM trial) [218]. However, the decision regarding the benefit of extended endocrine therapy beyond 5 years in ILC patients who are at risk for late recurrences depends on the genetic risk. The Breast Cancer Index is a genetic test which, in summary, gives information about whether the patient will benefit from extended endocrine therapy for a period from five to ten years and a prognostic result, which tells about the patient’s risk of having late recurrences [140]. The patients with high risk do seem to benefit from an extended endocrine therapy approach, but the population of patients where very rarely, there are discordance between the two results predicted by the breast cancer index test, there is paucity of data of the benefit of extended regimen in such cases. A recent study by Metzger et al. aimed to find the utility of BCI results in ILC vs. IDC cases and found that whereas the test identifies a lower percentage of cases of ILC as high risk for late recurrences, the ILC cases with high risk have a higher likelihood of obtaining benefits from an extended endocrine therapy regimen [219].

The decision concerning optimized endocrine therapy is determined by the menopausal status [220]. Whereas the preferred agent in postmenopausal women is aromatase inhibitors (AI), the data from the two phase 3 clinical trials, Tamoxifen and Exemestane Trial (TEXT) [221] and the Suppression of Ovarian Function Trial (SOFT) [222], show that the addition of ovarian function suppression (OFS) agents to an aromatase inhibitor has more benefits in terms of disease free survival in HR-positive breast cancers [223,224]. A recent single-center study by Record et al. studied the use of OFS in premenopausal women in ILC. They found that when adjusted to the age at diagnosis, a higher grade of disease seemed to benefit more from the addition of OFS to AI [225]. Even though the benefit of OFS+AI seems to be more in stage 2 and stage 3 disease, there is significant toxicity and poor tolerability associated with these agents. Another study called the POSITIVE trial [226] showed that stopping endocrine therapy in premenopausal women who wished to get pregnant and then restarting the therapy again after the pregnancy did not confer any substantial recurrence risk. However, long-term follow-up of this strategy is still to be explored.

### 9.5. Adjuvant Chemotherapy

Lobular carcinomas are known to have a quiescent histology with low mitotic rates; thus, the use of chemotherapy is limited in these tumors [227,228,229,230,231,232]. Two large meta-analyses by Gray et al. and Trapani et al., with around 38,000 cases each, also echo similar findings, with the former study showing no benefit of chemotherapy in ILC survival and the second finding no clear benefit of chemotherapy while comparing ILC and IDC-NST tumors [233,234]. However, adjuvant chemotherapy is effective in a subset of lobular carcinoma cases (gross lymph node involvement, larger tumor size, and lymphovascular invasion) [235]. Another separate study shows that patients with ILC with higher nodal involvement do benefit from anthracycline-based rather than treatment drugs [236].

### 9.6. HER-2 Targeted Therapies

Her-2 positivity ranges up to 3% in ILCs, with 15% of metastatic cases being Her-2 positive [237,238]. The efficacy of treatments like Trastuzumab in such cases is assumed to be similar to IDCs [239]. However, HR+ HER2+ early-stage ILC patients have been associated with late recurrence rates despite posttreatment with adjuvant trastuzumab and ET [240]. Though targeting HER-2 positive ILC tumors seems a little less beneficial with conventional Trastuzumab-based therapies, the MutHER and SUMMIT trials showed that combining Neratinib, a HER2-based tyrosine kinase inhibitor, with Fulvestrant led to better antitumor activity [91,241].

### 9.7. Immune-Checkpoint Inhibitor and Therapies

The utility of immune check inhibitors (ICIs) in ILCs has met with less success, and trials (GELATO Phase 2 trial, discussed in previous sections) exploring ICIs have been limited [148,242,243] and have also shown limited use in ILC cases. The results from Keynote-028 study in HR+/Her2 negative tumors had three cases with lobular histology, and two out of these three, which had stage 4 disease with metastasis, showed response to Pembrolizumab therapy [243]. Immune-rich ILC subset response to immunotherapy needs further exploration. The utility of ROS1 inhibitors and mTOR inhibitors has been discussed in previous sections.

### 9.8. Treatment Schema

The different modalities to treat ILC have been discussed above. However, the management strategies differ between early-stage ILC and metastatic ILC. Whereas in the early stage, the role of chemotherapy seems to be limited, as described in Section 9.1, the systemic treatment is endocrine therapy (Section 9.2) with or without chemotherapy. The decision to give chemotherapy is personalized to each patient depending on molecular risk, nodal status, stage of the disease, and menopausal status.

The first-line treatment in a metastatic setting is endocrine therapy with or without CDK4/6i, as shown by the MONARCH, MONALEESA, and PALOMA studies [210,214,215,216]. The recent SONIA TRIAL, however, challenged the use of CDK4/6i and proposed the approach of using ET alone initially, followed by CDK4/6i with ET [244]. The second-line therapy is important in metastatic patients owing to a higher incidence of mutations in these patients. Treatment options are based on mutations, viz., PIK3CA, which benefits from fulvesterant and targeted-based therapies (CAPITELLO 291 Trial) [82]. Other mutations need different other therapies (Appendix A). Additionally, in advanced cases, there can be the use of single chemotherapy. Also, with the knowledge of HER-2 low cases, especially in ILC, ADC can be plausible in the therapeutic approach, though larger studies are needed to confirm this. The information is summarized in Appendix A.

## 10. Important Clinical Trials Specific to ILC

Although we have tried to saliently mention the important aspects of clinical trials in the ILC domain in the respective sections, here we have tried to list and explain some ILC-specific trials which have high or promising translational importance:Trials researching the biology of ILC:LobularCard Trial: In this cross-sectional retrospective study, the population of interest is patients with LCIS and ILC. CDH1 is the unique mutation in ILCs; however, the germline mutation frequency is much less, and, hence, this trial aims to find out other genes associated with lobular breast cancer predisposition, using a panel of 113 genes in the “Illumina” protocol [115]. Thus, this will help in better understanding the disease, especially early-onset ILC.CDH1 Germline Mutations in Lobular Breast Cancer: Hereditary CDH1 germline mutations are associated with lobular carcinoma and associated hereditary diffuse gastric cancer. However, a few subsets of patients (<45 years) without hereditary diffuse gastric cancer also present with lobular cancer. Thus, this trial aims to investigate the prevalence of CDH1 in this specific population of women with early onset (<45 or <50) in situ or ILBC, bilateral LBC, or LBC with no family history of HDGC. Thus, this study might help in finding and better understanding the role of CDH1 as a susceptibility gene in lobular cancers [245].Drug intervention trials in ILCEarly-stage ILC:(a)Palbociclib and Endocrine Therapy for Lobular Breast Cancer Preoperative Study (PELOPS) [209]: CDK4/6 inhibitors plus endocrine therapy have shown promising results in HR+ breast cancer. In this randomized phase 2 trial studying ILC specifically, the patients are randomized to Tamoxifen versus Letrozole in the window phase, and the Ki67 score is measured in subsequent biopsies. The treatment phase includes patients who are randomized to tamoxifen plus palbociclib (CDK4/6 INHIBITOR) versus letrozole plus palbociclib.(b)Translational Breast Cancer Research Consortium 037 (TBCRC037) [246]: This is a randomized trial which studied neoadjuvant endocrine treatment strategies in postmenopausal woman with early-stage ILC. The study aims to find the efficacy of the most prevalent neoadjuvant therapies, viz., Tamoxifen, Anastrazole, and Fulvestrant, when given for a period of 21–24 days. The endpoint was measured by evaluating the Ki67 score.(c)Neoadjuvant Study of Targeting ROS1 in Combination with Endocrine Therapy in Invasive Lobular Carcinoma of the Breast (ROSALINE): Inhibitors of ROS 1 have been found effective in CDH1 tumors in preclinical studies (*synthetic lethality*) [73]. This trial is a neoadjuvant, single-arm, nonrandomized trial exploring the role of Entrectinib (ROS1 inhibitor) + letrozole in patients with early-stage ILC preoperatively, thus having the advantage of testing this treatment regimen in “treatment-naïve” tumors.Metastatic ILC:(a)MutHer II: In this single-arm multicohort phase 2 trial, the efficacy of Neratinib (irreversible pan-HER tyrosine kinase inhibitor) was evaluated in metastatic breast cancer patients with Her 2 mutations and not Her 2 amplifications. Although this trial was not ILC specific, they showed a 38% clinical benefit rate in fulvestrant-treated cases. Also, the clinical benefit rate was positively associated with ILC histology with Her 2 mutations, thus implying that this therapy may be more sensitive for such cases of ILC.(b)Crizotinib in Lobular Breast, Diffuse Gastric, and Triple Negative Lobular Breast Cancer or CDH1-mutated Solid Tumours (ROLo) [75]: This is a nonrandomized phase 2 study evaluating the role of a newer ROS1 inhibitor Crizotinib with fulvestrant in patients with E-cadherin defective, ER+ advanced, or metastatic lobular breast cancer.


## 11. Future Perspectives

In recent times, preclinical and clinical studies dedicated to ILC have increased and have displayed some interesting findings. However, the need of the hour is multicentric and multipopulation trials specific to ILC, which can unravel the biology of this disease in more detail.

ILC has been shown to have an adverse relation when there is higher expression of the bromodomain and extraterminal domain (BET) proteins BRD3/BRD4. Also, ILC is known to have FGFR-1 mutations, which also causes resistance to BET inhibition. Recent studies have found BET inhibition and FGFR inhibition to be a potential therapeutic strategy for endocrine-resistant cases [247]. Other promising potential therapies include studying the role of mTOR inhibitors since PI3K is one of the very important underlying pathways in ILC biology.

The tumor microenvironment in ILC needs to be dissected further with the use of newer spatial transcriptomic techniques since CAFs and macrophages seem to have an important role in the tumor microenvironment of ILC. Additionally, trials need to recruit more patients with immune-rich ILC to test the efficacy of anti-PDL1 therapies in greater detail. Gene signatures like LobSig need to be validated in different settings to further refine the robustness of ILC-specific gene signatures.

Finally, ILCs seem to have a significant number of cases which are now classified as Her 2 Low, and the use of antibody drug conjugates needs to be studied in these cases, preferably by designing multicentric randomized control trials.

## 12. Conclusions

In this review, we have provided a concise summary of significant translational findings relevant to invasive lobular carcinoma of the breast. Gaps in exploring this unique type of breast carcinoma are rapidly being amended in part due to exciting new technologies and high throughput assays. Future research will be focusing on a comprehensive understanding of the molecular, functional, and tissue microenvironment components of this unique tumor to offer a better personalized therapeutic approach to these patients.

## Figures and Tables

**Figure 1 cancers-15-05491-f001:**
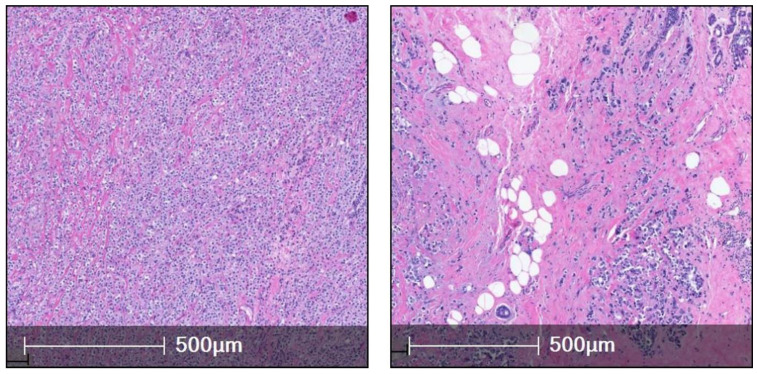
This figure (**left**, **right**) shows two cases of classic ILC (H&E).

**Figure 2 cancers-15-05491-f002:**
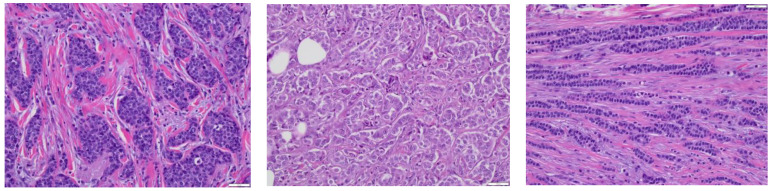
This figure shows alveolar ILC (**left**) (magnification 20×), solid ILC (**center**) (magnification 20×), and trabecular ILC (**right**) (magnification 20×).

**Figure 3 cancers-15-05491-f003:**
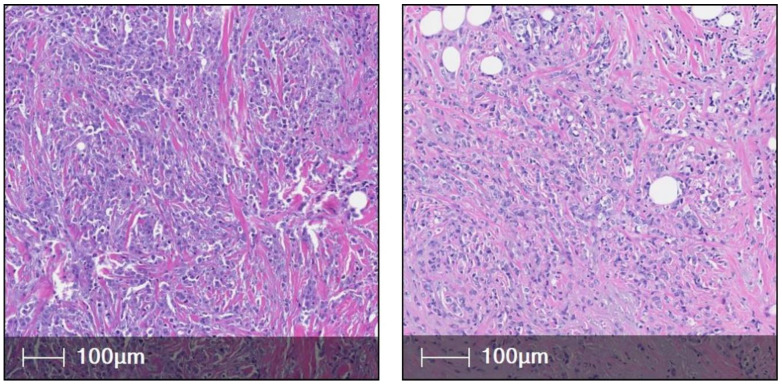
This figure (**left**, **right**) shows two cases of pleomorphic ILC (H&E).

**Figure 4 cancers-15-05491-f004:**
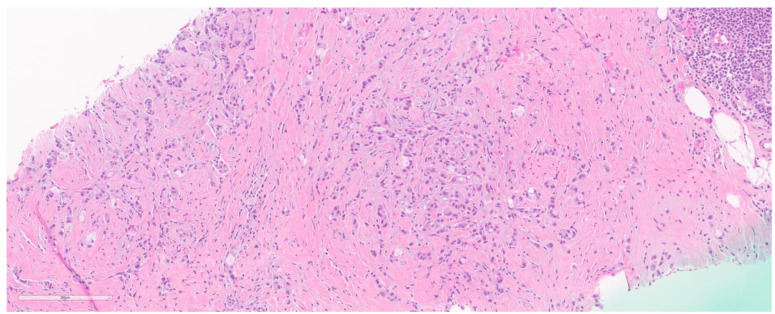
A case of histiocytoid ILC. Magnification (10×).

**Figure 5 cancers-15-05491-f005:**
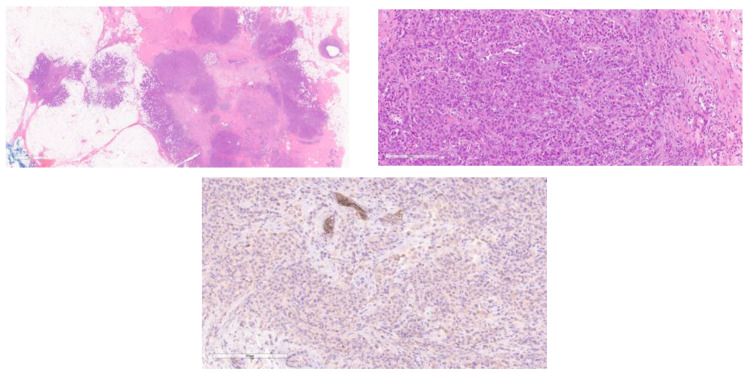
A case of ILC with signet ring cells. The (**top**) panel shows H&E images at low power (4×) (**left**) and high power (**right**) (magnification 20×) views. The (**bottom**) picture shows loss of E-cadherin expression.

**Figure 6 cancers-15-05491-f006:**
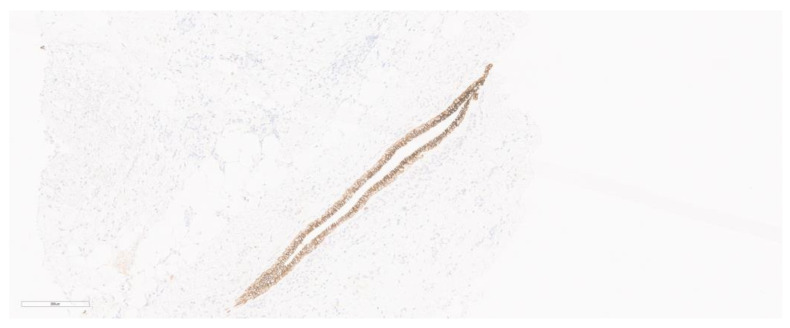
Loss of E-cadherin expression. Central normal duct showing E-cadherin positivity.

**Figure 7 cancers-15-05491-f007:**
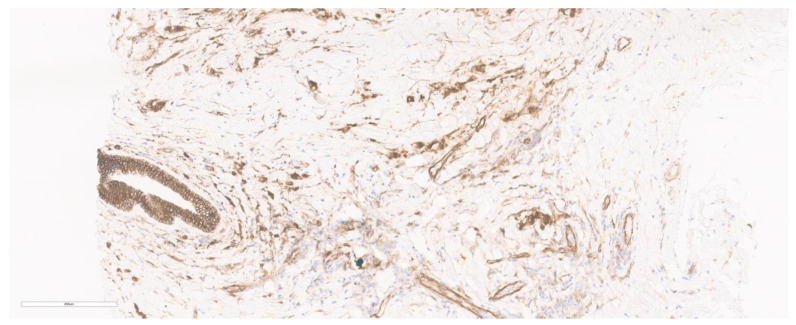
p120 IHC staining. The marked box shows cyto-membranous staining (p-120 IHC) of the tumor cells.

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
