# Peer review of "Lobular Carcinoma of the Breast: A Comprehensive Review with Translational Insights"

_cancers, 2023, doi:10.3390/cancers15225491_

Round 1

Reviewer 1 Report

Comments and Suggestions for Authors

This is a nice and well-illustrated review on invasive lobular carcinoma of the breast.

A few issues require to be further explored in review:

1. Differential diagnosis of histiocytoid and pleomorphic variants vs. invasive apocrine carcinoma; there may be a substantial morphologic and molecular (IHC) overlap, please refer to: Provenzano E, Gatalica Z, Vranic S. Carcinoma with apocrine differentiation. Breast Tumours. 5th ed. Lyon: IARC; 2019.

2. The frequency and the role of HER2 mutations in lobular carcinoma requires further elaboration, particularly in the metastatic setting. The same pertains to HER2-low prevalence and role in invasive lobular carcinoma.

3. The importance of anti-ER resistance is just briefly discussed. It requires further elaboration including the role of ER1 mutations in metastatic lobular carcinoma.

Reviewer 2 Report

Comments and Suggestions for Authors

In this article, the authors had a comprehensive review of the invasive lobular carcinoma, including the pathological and immunohistochemical characteristics, molecular mutation, gene expression profiling, microenvironment, radiology, as well as the treatment. Overall, the article is well organized and provides clinically useful information. Here are some comments and suggestions for improvement. 

  1. The authors may provide some clinical behaviors as well as risk factors of ILC.
  2. It is typically recognized that the ILC is primarily mediated by the loss of E-cadherin and evolves from lobular carcinoma in situ (LCIS). The authors may provide more information on the pathogenesis and the evolution.
  3. The pathological presentation of ILC may not be separately described with the immunohistochemical characteristics. For example, as the authors mention, Her2 overexpression of ILC is majorly seen in pleomorphic histology. 
  4. The authors only provide histological examples of classic ILC, pleomorphic, and Histiocytoid ILC. The authors may provide histological examples of other variants.
  5. Regarding the radiology aspect, the authors should provide some imaging features of ILC rather than only discussing the sensitivity of different kinds of imaging techniques.
  6. Regarding the treatment of ILC, more evidence-based data should be provided.
  7. The treatment of early-stage disease and metastatic setting may be discussed separately.
  8. Regarding the surgery, the author may also provide information on the axillary surgery.
  9. Regarding the endocrine therapy, is extended endocrine treatment be preferred in ILC patients? How does ovarian function suppression work in young ILC patients? And what’s the role of CDK4/6 inhibitors in the adjuvant treatment of ILC patients?
  10. The authors may provide some content related to neoadjuvant chemotherapy, as well as radiotherapy. 
  11. The authors may provide some information on novel treatments as well as new trials focusing specifically on the treatment of patients with ILC. Meanwhile, potential research direction might be discussed. 
  12. The authors may use tables to summarize information. Meanwhile, the paragraphing can be clearer. For example, the molecular alteration, the radiology aspect, as well as the Current Treatment Approaches may be described in a separate part.

Reviewer 3 Report

Comments and Suggestions for Authors

General comments
The spelling and punctuation are very good. No issues were detected.
Abstract
The abstract is concise. All the necessary information about the study is included.

Background
- The information provided in the introduction is important for the comprehension of the article.
- The objective of the study is clearly mentioned.
Methods
- The methods are sufficiently explained by the authors.

Results
- The results are presented in a very extensive way.
- The figures are really helpful and necessary for the completion of the authors' work.
Discussion
- The discussion is of great quality and includes updated data.
- The authors inform the reader about the study's limitations.
Conclusion
From the presented data, the conclusion is complete and represents the work that the authors did.

Revision

1) I would suggest at least adding one table presenting and summarizing the information of the 3.1 sections

2) Despite the major advances in breast cancer surgery, there are still numerous unanswered questions regarding the histological subtype of Invasive micropapillary carcinoma.

I would like a brief discussion on Invasive micropapillary carcinoma of the breast. Please consider citing the recently published articles:

https://pubmed.ncbi.nlm.nih.gov/35310681/

2) "HER2 is an established prognostic and predictive marker for patients with invasive breast cancer. The clinical and biological significance of HER2 overexpression in patients with ductal carcinoma in situ (DCIS) remains poorly defined. DCIS is a heterogeneous disease and some patients with DCIS will not progress to invasive breast cancer."

Add this important information and make a brief discussion on the clinical significance of HER2 expression in DCIS.

Consider citing recently published articles on this topic:

https://pubmed.ncbi.nlm.nih.gov/36352293/

Author Response

Thanks for reviewing our paper. We will work on point 1 and build a comprehensive table for the 3.1 section.

Regarding point 2, you mentioned 

"Despite the major advances in breast cancer surgery, there are still numerous unanswered questions regarding the histological subtype of Invasive micropapillary carcinoma. I would like a brief discussion on Invasive micropapillary carcinoma of the breast. Please consider citing the recently published articles: https://pubmed.ncbi.nlm.nih.gov/35310681/

Could you please clarify how this entity is related to lobular carcinoma and in which context you suggest including micropapillary carcinoma in our ILC review?

Regarding the second point 2, you mentioned 

2) "HER2 is an established prognostic and predictive marker for patients with invasive breast cancer. The clinical and biological significance of HER2 overexpression in patients with ductal carcinoma in situ (DCIS) remains poorly defined. DCIS is a heterogeneous disease and some patients with DCIS will not progress to invasive breast cancer." Add this important information and make a brief discussion on the clinical significance of HER2 expression in DCIS. Consider citing recently published articles on this topic:

https://pubmed.ncbi.nlm.nih.gov/36352293/.

It is not clear to us where do you suggest introducing DCIS Her2 expression in our review. If you can point out the line numbers, it will be of immense help.

Thanks in advance and looking froward to your answer.

Round 2

Reviewer 3 Report

Comments and Suggestions for Authors

well-written manuscript. it can be accepted without further revision